# Adiponectin receptor agonist AdipoRon improves skeletal muscle function in aged mice

Priya Balasubramanian[1†‡], Anne E Schaar[1†], Grace E Gustafson[1], Alex B Smith[1], Porsha R Howell[1], Angela Greenman[2], Scott Baum[3], Ricki J Colman[3,4], Dudley W Lamming[1,5], Gary M Diffee[2], Rozalyn M Anderson[1]*

[1]Department of Medicine, School of Medicine and Public Health, University of Wisconsin-Madison, Madison, United States; [2]Department of Kinesiology, University of Wisconsin-Madison, Madison, United States; [3]Wisconsin National Primate Research Center, University of Wisconsin-Madison, Madison, United States; [4]Department of Cell and Regenerative Biology, University of Wisconsin, Madison, United States; [5]Geriatric Research, Education, and Clinical Center, William S. Middleton Memorial Veterans Hospital, Madison, United States

*For correspondence:
rozalyn.anderson@wisc.edu

†These authors contributed equally to this work

Present address: ‡Center for Geroscience and Healthy Brain Aging, University of Oklahoma Health Sciences Center, Oklahoma, United States

Competing interest: The authors declare that no competing interests exist.

**Abstract** The loss of skeletal muscle function with age, known as sarcopenia, significantly reduces independence and quality of life and can have significant metabolic consequences. Although exercise is effective in treating sarcopenia it is not always a viable option clinically, and currently, there are no pharmacological therapeutic interventions for sarcopenia. Here, we show that chronic treatment with pan-adiponectin receptor agonist AdipoRon improved muscle function in male mice by a mechanism linked to skeletal muscle metabolism and tissue remodeling. In aged mice, 6 weeks of AdipoRon treatment improved skeletal muscle functional measures in vivo and ex vivo. Improvements were linked to changes in fiber type, including an enrichment of oxidative fibers, and an increase in mitochondrial activity. In young mice, 6 weeks of AdipoRon treatment improved contractile force and activated the energy-sensing kinase AMPK and the mitochondrial regulator PGC-1a (peroxisome proliferator-activated receptor gamma coactivator one alpha). In cultured cells, the AdipoRon induced stimulation of AMPK and PGC-1a was associated with increased mitochondrial membrane potential, reorganization of mitochondrial architecture, increased respiration, and increased ATP production. Furthermore, the ability of AdipoRon to stimulate AMPK and PGC1a was conserved in nonhuman primate cultured cells. These data show that AdipoRon is an effective agent for the prevention of sarcopenia in mice and indicate that its effects translate to primates, suggesting it may also be a suitable therapeutic for sarcopenia in clinical application.

## Editor's evaluation

In this manuscript, the authors provide promising results for the treatment of age-related sarcopenia with AdipoRon, a drug that targets the receptors for adiponectin. This is a well done study using an agonist (AdipoRon) involved in lipid and mitochondrial metabolism regulation to mitigate age-related muscle loss in mice.

## Introduction

Sarcopenia is the generalized and progressive decline in skeletal muscle mass that is accompanied by reduced muscle strength and reduced physical performance (*Cruz-Jentoft and Sayer, 2019*;

*Heymsfield et al., 2015*; *Santilli et al., 2014*). Loss of skeletal muscle function significantly increases the risk for impaired mobility, falls, fractures, and morbidity (*Angulo et al., 2016*; *Landi et al., 2013*). Sarcopenia also has significant metabolic consequences; in humans, skeletal muscle is the largest insulin-sensitive tissue in the body and its loss during aging dramatically increases the risk for sarcopenic obesity, diabetes, and cardiovascular disorders (*Cleasby et al., 2016*; *Koster et al., 2011*; *Srikanthan et al., 2010*; *Stenholm et al., 2008*). Exercise is the only known intervention to prevent sarcopenia (*Joseph et al., 2012*; *Law et al., 2016*; *Mcleod et al., 2019*; *Rebelo-Marques et al., 2018*); however, mobility disability and polymorbidity can prevent its effectiveness as a means to counter skeletal muscle functional loss. Pharmacological therapeutic interventions to treat or prevent sarcopenia would be a considerable clinical advance but none have yet been identified.

Skeletal muscle composition is matched to function such that the muscles required for posture are not equivalent to those engaged in physical activity. Contractile content includes individual muscle fibers with differing force and endurance properties and differing metabolic fuel preference. Skeletal muscle also includes non-contractile components, including satellite cells and stromal-derived stem cells, vascular tissues, adipose, and extracellular matrix. With increasing age, muscle fibers undergo atrophy, and the levels of fibrosis and intra-muscular fat increase within and around the fiber bundles (*Akazawa et al., 2017*; *Brioche et al., 2016*; *Lees et al., 2019*; *Mahdy, 2019*; *McGregor et al., 2014*). Sarcopenia is associated with loss of mitochondrial activity (*Andreux et al., 2018*; *Johnson et al., 2013*; *Petersen et al., 2015*; *Picca et al., 2019*), although in a variety of species from rodents to nonhuman primates and humans, it is the more oxidative fibers that are resilient to age-related atrophy (*Crupi et al., 2018*; *Lexell, 1995*; *Murgia et al., 2017*; *Pugh et al., 2013*). At the cellular level, aging is associated with fatty acid oxidation insufficiency and the accumulation of intracellular lipid (*Choi et al., 2016*; *Koves et al., 2008*; *Koves et al., 2013*). Importantly, our previous work in nonhuman primates show that age-related changes in mitochondrial energy metabolism, cellular redox, and lipid storage anticipate the onset of sarcopenia (*Pugh et al., 2013*), and occur in advance of physical activity decline and frailty (*Yamada et al., 2013*, *Yamada et al., 2018*). This suggests that changes in energy metabolism could be a contributing factor to age-related loss in muscle mass and function, and point to interventions that correct metabolic deficiency as potential agents for the treatment of sarcopenia.

Adiponectin is an adipose-derived peptide hormone that influences skeletal muscle cellular metabolism to activate lipid utilization and mitochondrial oxidative pathways (*Shetty et al., 2012*; *Stern et al., 2016*). Adiponectin signaling is transmitted via the ubiquitously expressed *Adipor1* and *Adipor2* receptors, with a poorly defined contribution from the T-cadherin receptor (*Parker-Duffen et al., 2013*; *Yamaguchi et al., 2007*). A major effector protein in adiponectin receptor activation in skeletal muscle is the AMP-activated protein kinase AMPK (*Iwabu et al., 2010*), a critical cellular energetic sensor (*Burkewitz et al., 2014*; *Hardie et al., 2012*). The actions of adiponectin somewhat resemble those of exercise: first, the adaptive response to exercise involves activation of metabolism in skeletal muscle and requires AMPK (*Fentz et al., 2015*; *Lee-Young et al., 2009*), and second, adiponectin and exercise share effectors downstream of AMPK including peroxisome proliferator-activated receptor gamma coactivator 1-alpha (PGC-1a), a key regulator of mitochondrial energy metabolism (*Miller et al., 2019a*) and a major player in the beneficial effects of exercise (*Cantó and Auwerx, 2010*; *Lira et al., 2010*; *Short et al., 2003*). In fact, genetic approaches to increase expression of PGC-1a in skeletal muscle specifically enhance endurance and protect against muscle aging in mice (*Garcia et al., 2018*; *Gill et al., 2018*). *Adipor1* knockout studies reveal that adiponectin signaling is required for maintenance of skeletal muscle oxidative fibers in vivo and for exercise capacity, with both AMPK and PGC-1a implicated in its actions (*Iwabu et al., 2010*). AdipoRon is a small-molecule adiponectin receptor agonist, and treatment with AdipoRon corrects metabolic dysfunction in genetically obese and metabolically dysfunctional db/db mice (*Okada-Iwabu et al., 2013*). Although AdipoRon has been shown to impact gene expression in skeletal muscle in short-term treatment of young mice, the ability of the drug to impact skeletal muscle in older mice is unknown.

# Results

## AdipoRon improves skeletal function and indices of insulin sensitivity in aged mice

To assess the ability of AdipoRon to influence skeletal muscle aging, AdipoRon (1.2 mg/kg) or vehicle (PBS) were administered intravenously to 25-month-old C57BL/6J male mice three times per week for 6 weeks. AdipoRon treatment resulted in a decline in body weight in aged animals, which did not reach statistical significance (*Figure 1A*). Body composition analysis revealed significant loss in fat mass in AdipoRon treated animals, although there was no difference in lean mass compared to Controls (*Figure 1B–C*). AdipoRon treated animals had significantly lower fasting glucose levels (*Figure 1D*) and a trend toward decreased fasting insulin levels (*Figure 1E*). In line with these findings, the HOMA-IR index of insulin resistance was significantly lower in AdipoRon treated animals. Circulating levels of triglycerides were not different between Control and AdipoRon treated animals. In metabolic chamber assessments, food consumption, spontaneous activity, energy expenditure (EE), or respiratory exchange ratio (RER) did not differ significantly between Control and AdipoRon treated animals over a 24-hr period (*Figure 1F*; *Figure 1—figure supplement 1–S1*). Parsing these data into smaller increments revealed differences in EE and RER at discrete time points only (t-test; unadjusted p<0.05). In gastrocnemius, a mixed fiber muscle in the lower leg with Type II fibers predominating, PGC-1a protein levels were higher from AdipoRon treated mice than from controls (*Figure 1G*), suggesting that activation of PGC-1a could be mediating the effects of AdipoRon in aged mice. Levels of VDAC (voltage-dependent anion channel), an index of mitochondrial content, were not different between control and AdipoRon treated mice, indicating that mitochondrial biogenesis was not induced by AdipoRon in gastrocnemius.

The therapeutic potential of AdipoRon was evaluated via in vivo and ex vivo functional measures. Baseline measures for rotarod did not differ between the groups; however, after 6 weeks of treatment, the AdipoRon treated animals tended to spend more time on the rotarod (p=0.08) suggesting a trend toward improved muscle function and agility (*Figure 1H*). Ex vivo contractility measurements in extensor digitorum longus (EDL, predominantly glycolytic) and soleus (predominantly oxidative) muscles were performed to determine the effects of AdipoRon treatment on whole muscle contractile behavior. Muscle maximal specific force generation and fatigability are known to be dependent on the type and quantity of expression of myosin heavy chains isoforms (*Geiger et al., 2000*). Consistent with this, the decline of tetanic force after initial simulation (fatigue) observed in Type II fiber-enriched EDL was greater than that of Type I enriched soleus muscle in untreated aged mice. Force generation after repeated tetanus stimulation was significantly greater in the EDL from AdipoRon treated animals compared to controls, signifying a greater resistance to fatigue (*Figure 1I*). Electrical stimulation-induced force generation after a 20-min recovery period was also significantly higher, indicative of greater muscle endurance following AdipoRon treatment. Interestingly, the beneficial effects of AdipoRon on fatigability were detected in EDL but not soleus muscle that had numerically greater values for both parameters at baseline compared to EDL (*Figure 1J*). These data suggest that oxidative muscles (soleus) are less sensitive to AdipoRon treatment compared to muscles with more reliance on glycolytic metabolism (EDL). PGC-1a is a key regulator of fiber type switching (*Lin et al., 2002*) and is engaged downstream of adiponectin receptor signaling (*Iwabu et al., 2010*).

## AdipoRon changes muscle composition including fiber-type-specific metabolic activation

A change in muscle fiber type distribution within muscle is an outcome of chronic exercise (*Mishra et al., 2015*; *Yan et al., 2011*), where composition is adapted to functional demand. Differences in fiber type distribution within muscle are associated with distinct contractile properties (*Zierath and Hawley, 2004*). Gastrocnemius muscle was analyzed as representative of a *predominantly* glycolytic muscle (Type IIb and Type IIx muscle fibers). Immunohistological detection of myosin isoforms in gastrocnemius muscle cross-section (*Figure 2A*) revealed a significantly greater proportion of Type IIa myosin expressing fibers in muscle from AdipoRon treated mice compared to untreated control mice (*Figure 2B*). Histochemical detection of cytochrome c oxidase activity revealed an increase in intensity of COXIV activity staining in Type IIb fibers (*Figure 2C*). Interestingly, AdipoRon treatment was associated with a very small decrease in fiber size in Type IIb fibers (*Figure 2D*), indicating there may

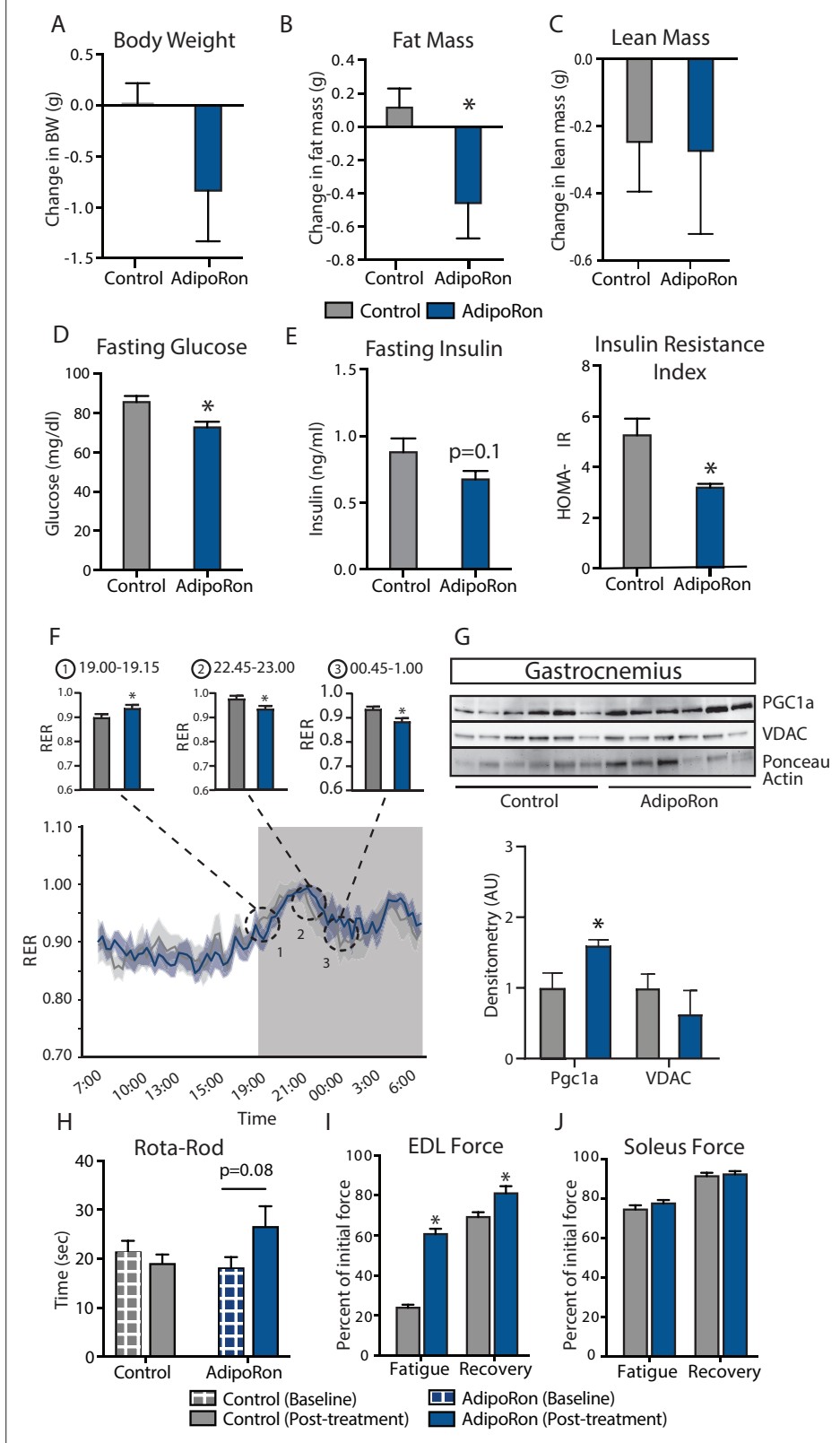

**Figure 1.** Metabolic and functional effects of chronic AdipoRon treatment in aged mice. Chronic AdipoRon treatment (1.2 mg/kg BW, intravenous injection three times per week for 6 weeks) in aged male C57BL/6J mice (25 months old). (**A**) Change in body weight, (**B**) change in fat mass, and (**C**) lean mass. (**D**) Fasting glucose, (**E**) fasting insulin and HOMA-IR (Control [n=10] and AdipoRon [n=8]), (**F**) mean respiratory exchange ratios

*Figure 1 continued on next page*

*Figure 1 continued*

(RERs) showing confidence intervals during 24 hr in metabolic chambers, box signifies night (Control [n=10] and AdipoRon [n=8]), (**G**) immunodetection of mitochondrial marker VDAC and PGC1a protein expression in gastrocnemius (Control [n=6] and AdipoRon [n=6]), (**H**) in vivo latency to fall on Rota-Rod testing (Control [n=10] and AdipoRon [n=7]). Functional muscle changes assessed by measuring peak force after tetanic stimulation in ex vivo contractility experiments in isolated (**I**) extensor digitorum longus (EDL) and (**J**) soleus (Control [n=10] and AdipoRon [n=8]). Data shown as average ± SEM, significance determined by Student's t-test (*p<0.05).

The online version of this article includes the following figure supplement(s) for figure 1:

**Figure supplement 1.** Metabolic effects of chronic AdipoRon treatment.

be structural adaptation to changes in mitochondrial activation status. Quantitative PCR confirmed Type IIb myosin isoform was by far the dominant fiber type in gastrocnemius. AdipoRon treated mice had a modest reduction in expression of all myosin isoforms compared to controls although none of the differences were significant, and overall fiber type distribution was not altered (*Figure 2E*). Soleus muscle is a predominantly oxidative muscle, and RT-PCR confirmed enrichment for Type I and Type IIa myosin isoforms (*Figure 2—figure supplement 1*). Although functional differences were not detected ex vivo in soleus, AdipoRon treatment significantly increased expression of Type I and Type IIx myosin, and numerically increased expression of Type IIa and Type IIb, although here too the overall distribution of fiber types was not altered (*Figure 2—figure supplement 1*).

To investigate potential feedback from AdipoRon treatment, transcriptional expression of adiponectin receptors *Adipor1* and *Adipor2* was measured relative to the expression of *Rn18s* ribosomal RNA. In gastrocnemius muscle, levels of *Adipor1* were significantly greater than *Adipor2* (*Figure 2F*). Levels of both receptors were higher in old mice compared to young mice, corroborating previous reports that adiponectin receptor expression changes with age (*Ito et al., 2018*). Expression levels of both receptors were numerically lower in AdipoRon treated mice and not significantly different from the levels in young mice. In soleus muscle, *Adipor1* levels were higher than those of *Adipor2,* but the expression was not altered with age (*Figure 2—figure supplement 1*). Although levels for each receptor were numerically higher in AdipoRon treated mice, neither change reached significance. The known age-associated increase in fibrosis in muscle (*Wood et al., 2014*) prompted an investigation of the effects of AdipoRon treatment on collagen content in aged animals. Histochemical staining for collagen showed that over this relatively short treatment period AdipoRon did not impact collagen content or fibrosis in the aged muscle (*Figure 2—figure supplement 2*). Taken together, these data show that chronic AdipoRon treatment mimics exercise including changes in fiber type composition with more oxidative fibers, a shift in glycolytic muscle fibers toward a more oxidative phenotype, and a correction of age-related elevation of adiponectin receptor expression.

## AdipoRon improves contractile force in young mice in a muscle-type-specific manner

To assess the chronic effects of AdipoRon on metabolism and muscle function in young mice, we administered AdipoRon (1.2 mg/kg) or vehicle (PBS) three times per week for 6 weeks by tail vein injections beginning at 2 months of age in male B6C3F1 hybrid mice. Chronic AdipoRon treatment did not alter body weight (*Figure 3A*) fasting glucose, nor insulin levels in young animals (*Figure 3B–C*). Ex vivo contractility measurements were performed in EDL and soleus muscles from the young Control and AdipoRon treated mice. Chronic AdipoRon treatment resulted in significantly improved ability to recover after repeated tetanus stimulation in EDL muscle suggesting higher resistance to fatigue (*Figure 3D*). Consistent with results from the aged mice, the effect of AdipoRon was muscle-type specific as it did not impact measures of contractility and force in the soleus muscle (*Figure 3E*). This confirms that AdipoRon preferentially improves function in fast muscle with predominantly Type IIb/IIx fibers independent of the age of the mice during treatment. PGC-1a protein levels were higher in gastrocnemius from AdipoRon treated mice than from controls (*Figure 3F*). Total transcript levels of PGC1a were higher in gastrocnemius from AdipoRon treated mice but not in soleus (*Figure 3G*), although transcript levels of selected PGC1a target genes (*Cpt1a*, *Acox1*, and *Acadm*) were not altered by AdipoRon treatment. The *Ppargc1* gene produces several isoforms including PGC-1a1 and PGC-1a4 from the canonical promoter and variants PGC-1a2, and PGC-1a3 from an alternate

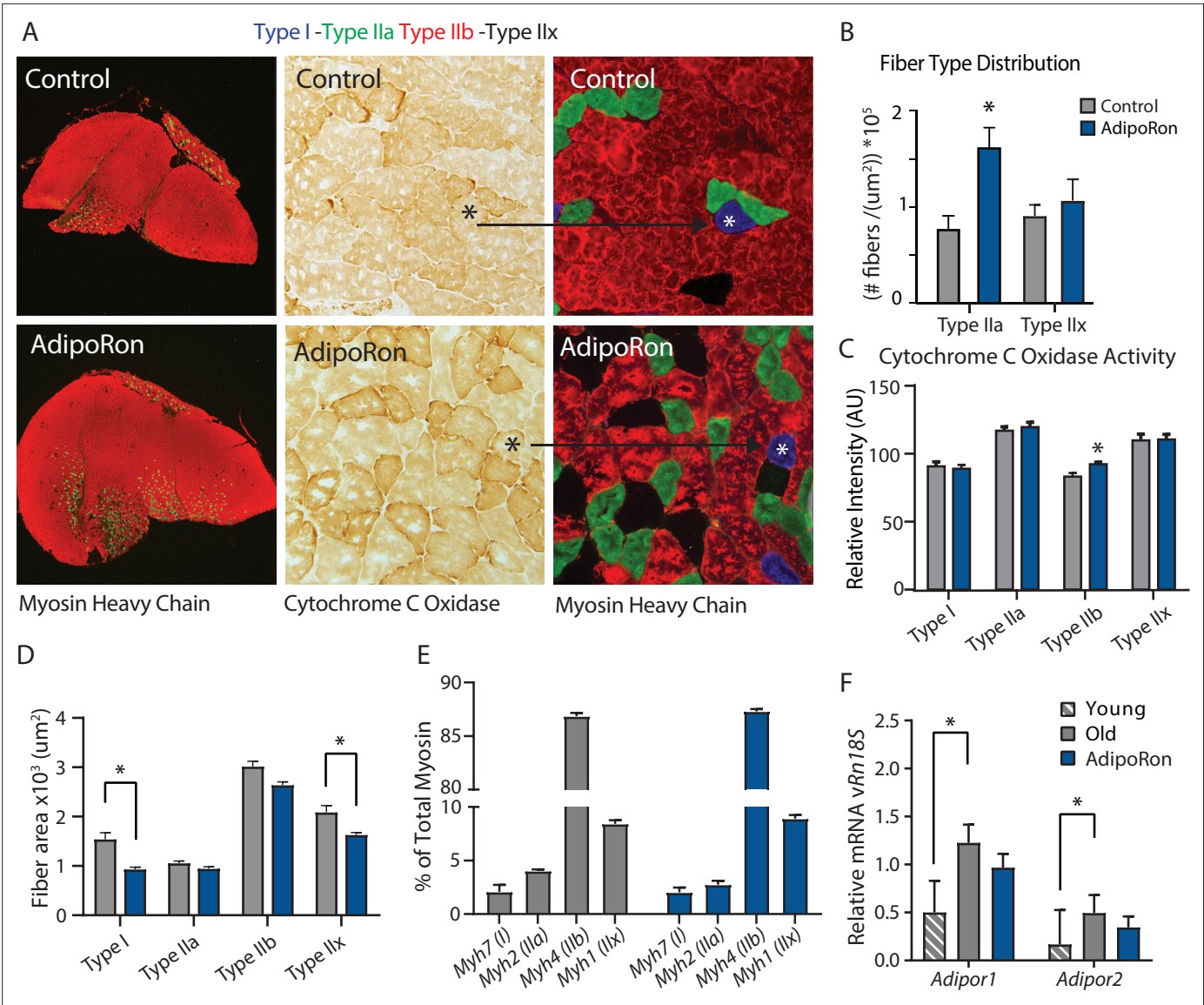

**Figure 2.** Fiber type-specific metabolic and structural changes in response to AdipoRon. Chronic AdipoRon treatment in aged male C57BL/6J mice (25 months old). (**A**) Immunodetection of Type IIa (green) and Type IIb (red) shown at 2.5× magnification (Left), representative histochemical staining for Cytochrome C Oxidase activity (Middle), and immunohistochemistry for Type I (blue), Type IIa (green), Type IIb (red), and Type IIx (black) in adjacent gastrocnemius muscle sections at 20× magnification (Right). (**B**) Type IIa and Type IIx fiber counts normalized to (**C**) Cytochrome C Oxidase staining intensity analysis (**D**) and cross-sectional area of individual fibers. Data shown as mean ± SEM, significance determined by t-test (*p<0.05). (**E**) mRNA expression as percent of total myosin heavy chain isoforms (Control [n=10] and AdipoRon [n=8]). (**F**) Relative mRNA expression (relative to *Rn18s* RNA) of adiponectin receptors *Adipor1* and *Adipor2* from young (Control [n=6] and AdipoRon [n=6]) and aged (Control [n=10] and AdipoRon [n=8]) mice. Data shown as fold change of average difference from *Rn18s* ± SEM, significance determined by ANOVA (*p<0.05).

The online version of this article includes the following figure supplement(s) for figure 2:

**Figure supplement 1.** Chronic AdipoRon treatment in aged mice soleus.

**Figure supplement 2.** Effect of AdipoRon treatment on aged muscle fibrosis.

promoter (*Ruas et al., 2012*). AdipoRon altered the profile of PGC-1a isoforms indicating a change in promoter preference .

To determine whether AdipoRon mediated changes in muscle function were coincident with changes in muscle composition, tissue sections from young mice were subject to immunodetection of myosin heavy chain isoforms. No changes in fiber cross-sectional area were identified, presumably because young mice are in peak physical condition (*Figure 3—figure supplement 1*). Histochemical

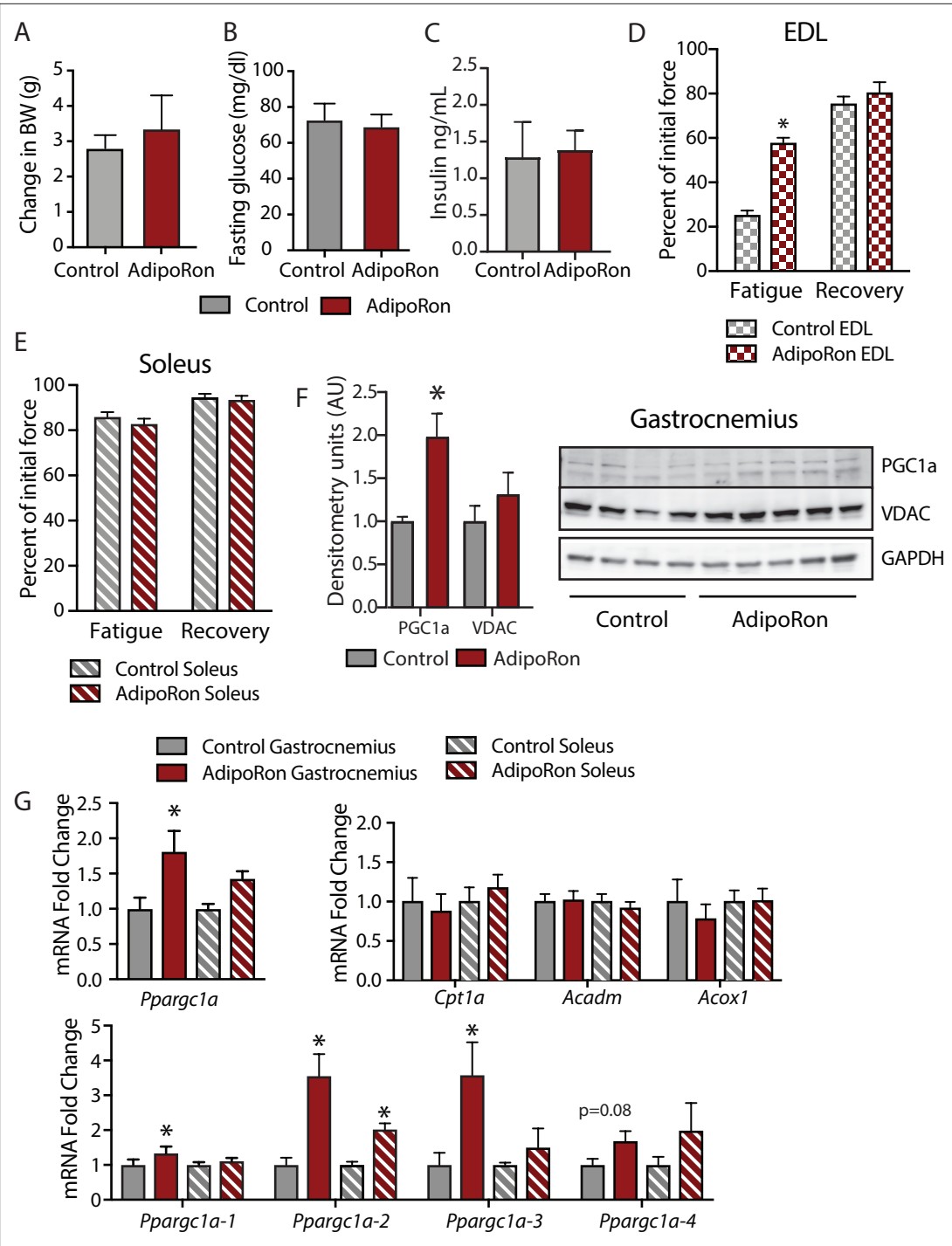

**Figure 3.** Metabolic and functional effects of chronic AdipoRon treatment in young mice. Chronic AdipoRon treatment (1.2 mg/kg BW, intravenous injection three times per week for 6 weeks) in young male B6C3F1 mice (3.5 months). Measures of (**A**) body weight, (**B**) fasting glucose, (**C**) fasting insulin (Control [n=5] and AdipoRon [n=5]). Measures of peak force after tetanic stimulation in ex vivo contractility experiments in isolated EDL (**D**) and Soleus (**E**) muscles. (**F**) Immunodetection of protein levels of PGC1a and VDAC in gastrocnemius muscle. (**G**) mRNA expression of *Ppargc1a*, PGC-1a gene targets, and *Ppargc1a* isoforms in gastrocnemius and soleus muscle. Data shown as mean ± SEM, significance determined by Student's t-test (*p<0.05). EDL, extensor digitorum longus.

The online version of this article includes the following figure supplement(s) for figure 3:

**Figure supplement 1.** Effects of chronic AdipoRon treatment in young adult mice.

detection of maximal cytochrome c oxidase (Complex IV, COXIV) and succinate dehydrogenase (Complex II, SDH) activity (*Mahad et al., 2009*), similarly showed no differences in young mice with or without AdipoRon treatment (*Figure 3—figure supplement 1*). Fiber area measurements revealed that the fiber size was inversely proportional to mitochondrial activity (Type IIb>IIx>I>IIa), which has been described previously as the muscle fiber type-fiber size paradox (*van Wessel et al., 2010*).

## Acute treatment with AdipoRon activates PGC-1a in a muscle-type-specific manner

To understand the differences among muscle groups in terms of PGC-1a expression and adiponectin receptor distribution, gene expression analysis was conducted in extracts from gastrocnemius and soleus muscles from young male B6C3F1 hybrid mice (6 weeks old). Endogenous levels of *Ppargc1a* expression were significantly higher in soleus than gastrocnemius muscle (*Figure 4A*), which is to be expected due to the higher dependence of soleus muscle on oxidative metabolism (*Koves et al., 2005*). Interestingly, the distribution of isoforms was largely equivalent between the two muscle types, with *Ppargc1a1* being the dominant isoform, lesser contributions from *Ppgargc-1a4* and *Ppargc1a2*, and lowest levels detected for *Ppargc1a3* (*Figure 4B*). The possibility that differences in responsivity to AdipoRon between the gastrocnemius and soleus muscle types observed in the chronic treatment studies could be due to differences in adiponectin receptor expression was investigated. Gene expression analysis revealed that in young mice, there was no overt difference in the expression levels of *Adipor1* and *Adipor2* between gastrocnemius and soleus, though *AdipoR1* expression was notably higher than *Adipor2* for both muscle types (*Figure 4C*).

To investigate the early response to AdipoRon in skeletal muscle, the impact of a single intravenous injection of AdipoRon (1.2 mg/kg BW) was investigated in muscles from young male B6C3F1 hybrid mice (6 weeks old) collected 90 min postinjection. AdipoRon treatment increased the gene expression of *Ppargc1a* in gastrocnemius but not soleus muscle (*Figure 4D*). Exercise training has been reported to induce the expression of the canonical promoter-driven *Ppargc1a1* and *Ppargc1a4*, with a lesser effect on isoforms expressed from the alternate promoter. Acute treatment with AdipoRon resulted in an upward trend of transcript levels of most highly expressed isoform *Ppargc1a1*, a significant increase in the *Ppargc1a4* isoform in gastrocnemius muscle, although increases in expression of *Ppargc1a2* and *Ppargc1a3* did not reach significance (*Figure 4E*). No changes were seen with the expression of *Ppargc1a* isoforms in soleus muscle, and gene targets of PGC-1a involved in fatty acid metabolism (*Cpt1a, Acox1*, and *Acadm*) were increased in response to acute AdipoRon treatment in gastrocnemius but not in soleus muscle (*Figure 4F*). Exercise training has also been linked to changes in adiponectin receptor expression, where an 8-week exercise regimen increased bicep femoris muscle mRNA expression of *Adipor1* but had no effect of *Adipor2* (*Huang et al., 2006*). Within our study, AdipoRon similarly increased the expression of *Adipor1* in gastrocnemius muscle but had no effect on levels of *Adipor2* (*Figure 4G*). AdipoRon had no effect on the expression levels of either adiponectin receptor in the soleus muscle, indicating that the effects of AdipoRon are muscle-type specific.

## AdipoRon stimulates mitochondrial function in cultured fibroblasts

Adiponectin receptor stimulation by AdipoRon has been linked to activation of AMPK and PPARa (peroxisome proliferator-activated receptor alpha) pathways in muscle and liver, including increased PGC-1a transcription and activation of expression of its gene targets (*Okada-Iwabu et al., 2013*). Studies of renal function in a genetic model of diabetes (db/db) show AMPK and PPARa engagement by AdipoRon and have revealed changes in lipid parameters in response to AdipoRon in renal endothelial cells (*Choi et al., 2018*). Less is known of the functional consequence of PGC-1a activation by AdipoRon as it relates to cellular metabolism and mitochondrial function. Cultured NIH-3T3 fibroblasts were treated for 48 hr with 50 μm AdipoRon treatment. Mitochondrial membrane potential measured by TMRE assay was increased after 2 and 4 hr of 50 μm AdipoRon treatment (*Figure 5A*). The impact of AdipoRon on mitochondrial function was coincident with an increase in cellular oxygen consumption measured over the same time period (*Figure 5B*). Cellular ATP concentration was initially lower at 2 and 4 hr following AdipoRon 50 μm treatment, but subsequently increased and was significantly higher 12 hr following treatment (*Figure 5C*). Adaptive responses to metabolism include possible effects on cellular growth and remodeling of the mitochondrial network. The doubling time of NIH-3T3 is approximately 24 hr and our previous studies have shown that increased respiration is

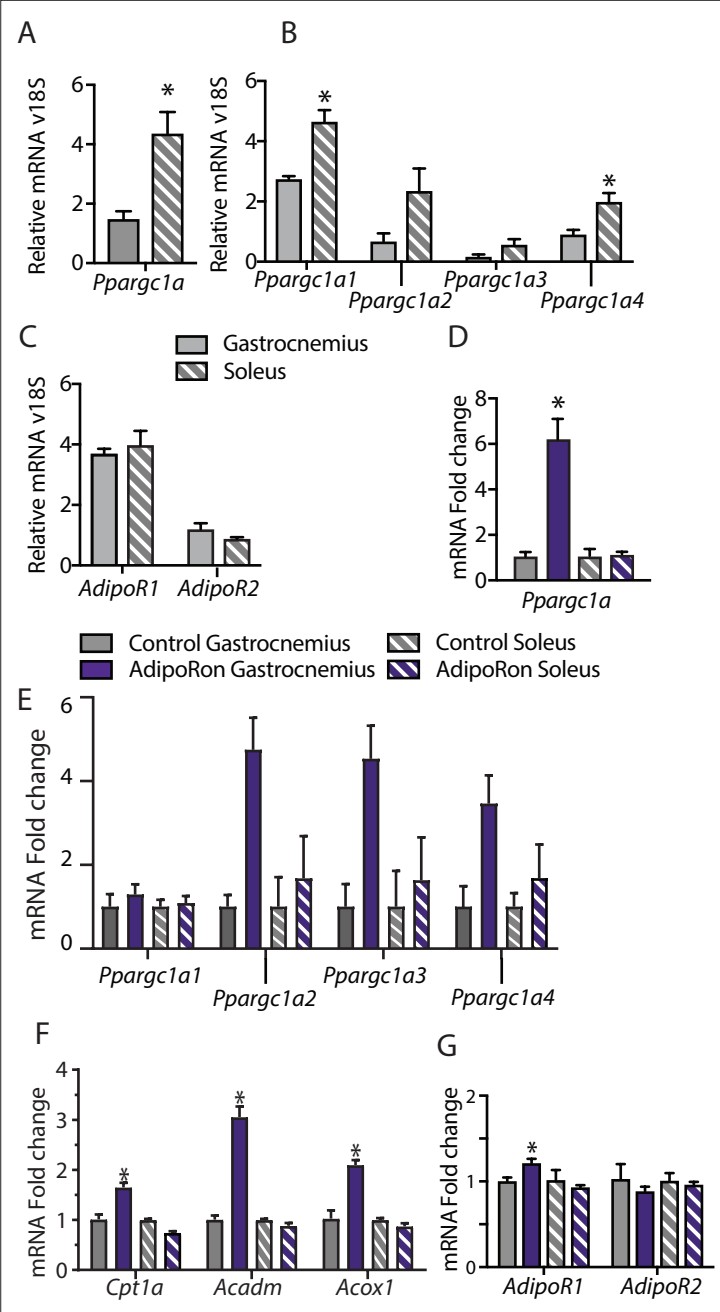

**Figure 4.** Muscle-type-specific transcriptional response to acute AdipoRon treatment. Comparison between gastrocnemius and soleus muscles of (**A**) *Ppargc1a*, (**B**) *Ppargc1a* isoforms, and (**C**) adiponectin receptors *Adipor1* and *Adipor2* mRNA relative to *Rn18s* (Gastrocnemius [n=3] and Soleus [n=3]). Fold changes in gene expression in gastrocnemius and soleus muscle of 6-week-old male B6C3F1 hybrid mice 90 min post-IV injection with AdipoRon (1.2 mg/kg BW) (**D**) *Ppargc1a*, (**E**) *Ppargc1a* isoforms, (**F**) gene targets of PGC1a, and (**G**) adiponectin receptors R1 and R2 (Control [n=3] and AdipoRon [n=5]). Data shown as mean ± SEM, significance determined by Student's t-test (*p<0.05).

accompanied by slower growth in cultured fibroblasts (**Miller et al., 2019b**). Consistent with this, cell proliferation was negatively impacted by 24 hr exposure to AdipoRon regardless of seeding density (**Figure 5D**). We had previously shown that mitochondrial organization is influenced by chronic elevation of PGC-1a (**Miller et al., 2019b**). Mitochondria were visualized in fixed fibroblasts by immuno-fluorescence detection of TOMM20 following 24 hr AdipoRon treatment (**Figure 5E**). Treated cells showed quantitatively significant differences in fluorescent intensity and in mitochondrial network

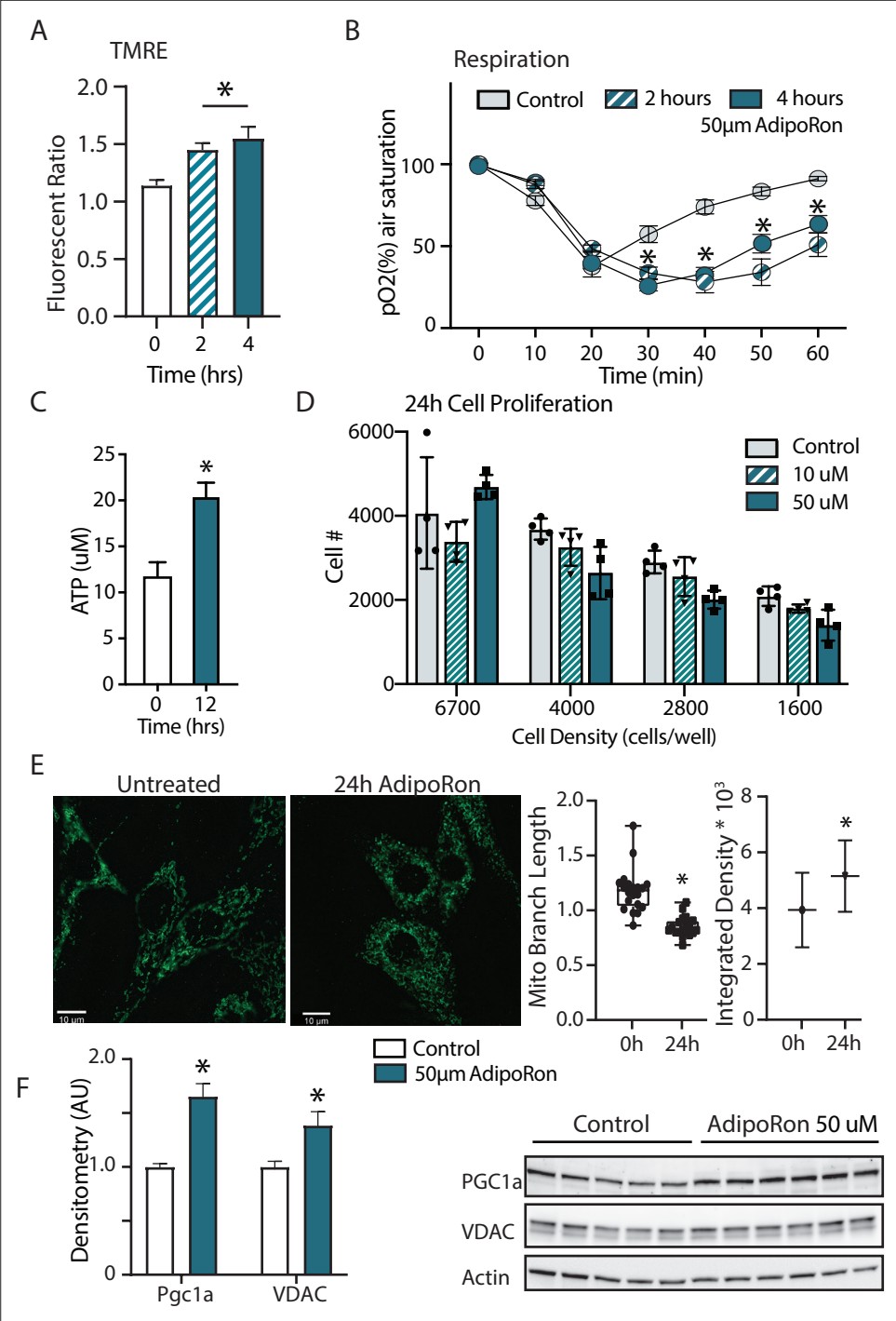

**Figure 5.** Short term impact of AdipoRon treatment on mitochondrial metabolism in murine fibroblasts. (**A**) Mitochondrial membrane potential measured by TMRE assay at 2 and 4 hr of AdipoRon treatment (n=6 per group), (**B**) oxygen consumption measured in response to AdipoRon treatment (Control-open circle, AdipoRon 2 hr-hatched circle, AdipoRon 4 hr-filled circle) by oxoplate assay (n=5 per group), (**C**) ATP concentration after indicated times with 50 μm AdipoRon treatment, (**D**) impact of 24 hr of AdipoRon treatment on cell proliferation of differing seeding densities as assessed by CyQUANT assay (n=4 per group), (**E**) fluorescent detection of mitochondria in fixed NIH-3T3 following AdipoRon treatment including quantitation of integrated density (product of mean intensity and mitochondrial area), and mean mitochondrial skeleton branch length (15–20 cells per time point), (**F**) Western blot showing protein levels of PGC1a and VDAC after 48 hr of 50 μm AdipoRon treatment of NIH-3T3 fibroblasts (n=5–6 per group), data shown as average shown as mean ± SEM (*p<0.05 via Student's t-test).

*Figure 5 continued on next page*

*Figure 5 continued*

The online version of this article includes the following figure supplement(s) for figure 5:

**Figure supplement 1.** Mitochondrial morphology analysis.

architecture as mitochondria tended toward shorter branch morphologies. Finally, immunodetection of VDAC (voltage-dependent anion channel) revealed a modest but significant increase in VDAC, an index of mitochondrial content, matching the immunofluorescence data (*Figure 5F*). A significant increase in PGC-1a protein was also detected. These data reveal prominent early and adaptive effects of acute AdipoRon treatment on the mitochondrial regulator PGC-1a and on mitochondrial function and organization and link these PGC-1a-associated metabolic changes to delayed proliferation and growth.

## AdipoRon actions are conserved among cell types in culture

In cultured muscle cells, AdipoRon activates AMPK and increases the expression of PGC1a and its gene targets (*Ito et al., 2018*; *Okada-Iwabu et al., 2013*). The degree to which AdipoRon action is conserved among other cell types and among species remains an open question. *Adipor1* and *Adipor2* are expressed in diverse cell types, including murine NIH-3T3, differentiated mouse myoblast C2C12 cells, and nonhuman primate peripheral blood mononuclear cells (PBMCs), although there is cell type specificity in levels and relative ratios of the two adiponectin receptors (*Figure 6A*). Our data revealed functional changes in mitochondria in response to AdipoRon treatment in NIH-3T3 fibroblasts and we sought to investigate whether the molecular underpinnings for these observations were linked to AMPK and to PGC-1a. NIH-3T3 fibroblasts were treated with AdipoRon treatment (10 or 50 µm) for 10 min. Increased levels of activating phosphorylation of AMPK at Thr172 were detected in extracts of AdipoRon treated cells compared to controls for both the lower and higher doses (*Figure 6B*). These data align with prior reports of AdipoRon directed AMPK activation in C2C12 myoblasts and glomerular endothelial cells (*Choi et al., 2018*; *Ito et al., 2018*; *Okada-Iwabu et al., 2013*). Increased levels of *Ppargc1a* transcript were detected after 90 min of AdipoRon treatment at 50 µm but not 10 µm concentration (*Figure 6C*). Although the transcript of PGC-1a itself was not increased at the lower dose, expression levels of known targets of PGC-1a (*Cpt1a, Acox1*, and *Acadm*) were significantly increased at 10 µm and at 50 µm, indicating that PGC-1a activation at the lower dose of AdipoRon may occur at the posttranscriptional level. On day 7, differentiated C2C12 myotubes AdipoRon significantly induced phosphorylation of AMPK at Thr172 (*Figure 6D*). *Ppargc1a* gene expression was induced at the higher dose only, and PGC-1a target Acox1, but not Cpt1 or Acadm, was also induced at the 50 µm dose (*Figure 6E*). Mitochondrial membrane potential was also increased in response to AdipoRon in the myotubes (*Figure 6F*). Myosin isoform profiling revealed enriched expression of isoforms that are primarily associated with an oxidative phenotype that in vivo is less responsive to AdipoRon treatment (*Figure 6G*). The response to AdipoRon was next investigated in a diverse cell population derived from nonhuman primates. PBMCs were isolated from blood taken from adult male rhesus monkeys. The cells were cultured overnight and subsequently treated with AdipoRon. Activating phosphorylation of AMPK at Thr172 was increased after 10 min of AdipoRon treatment at 50 µm but not 10 µm concentration (*Figure 6H*). One hour following AdipoRon treatment of PBMCs significant increases in transcript levels of *Ppargc1a* and PGC-1a gene targets were detected at both the lower and higher doses (*Figure 6I*). These data demonstrate that the molecular response to AdipoRon is conserved among cell types, although there is cell type specificity in the expression of adiponectin receptors and in the sensitivity of response through AMPK signaling and activation of PGC-1a transcriptional co-activation.

## Discussion

We have provided evidence of the therapeutic potential of AdipoRon for the treatment of sarcopenia in mice. In vivo and ex vivo, functional assessments of skeletal muscle revealed that 6 weeks of AdipoRon treatment improved skeletal muscle function in aged mice. Interestingly, the effect was muscle type specific, where AdipoRon impacted functional, metabolic, and gene expression outcomes in *predominantly* glycolytic muscles (EDL and gastrocnemius) but not in *predominantly* oxidative soleus muscle. Skeletal muscle is known for its heterogeneous population of muscle fibers, each with

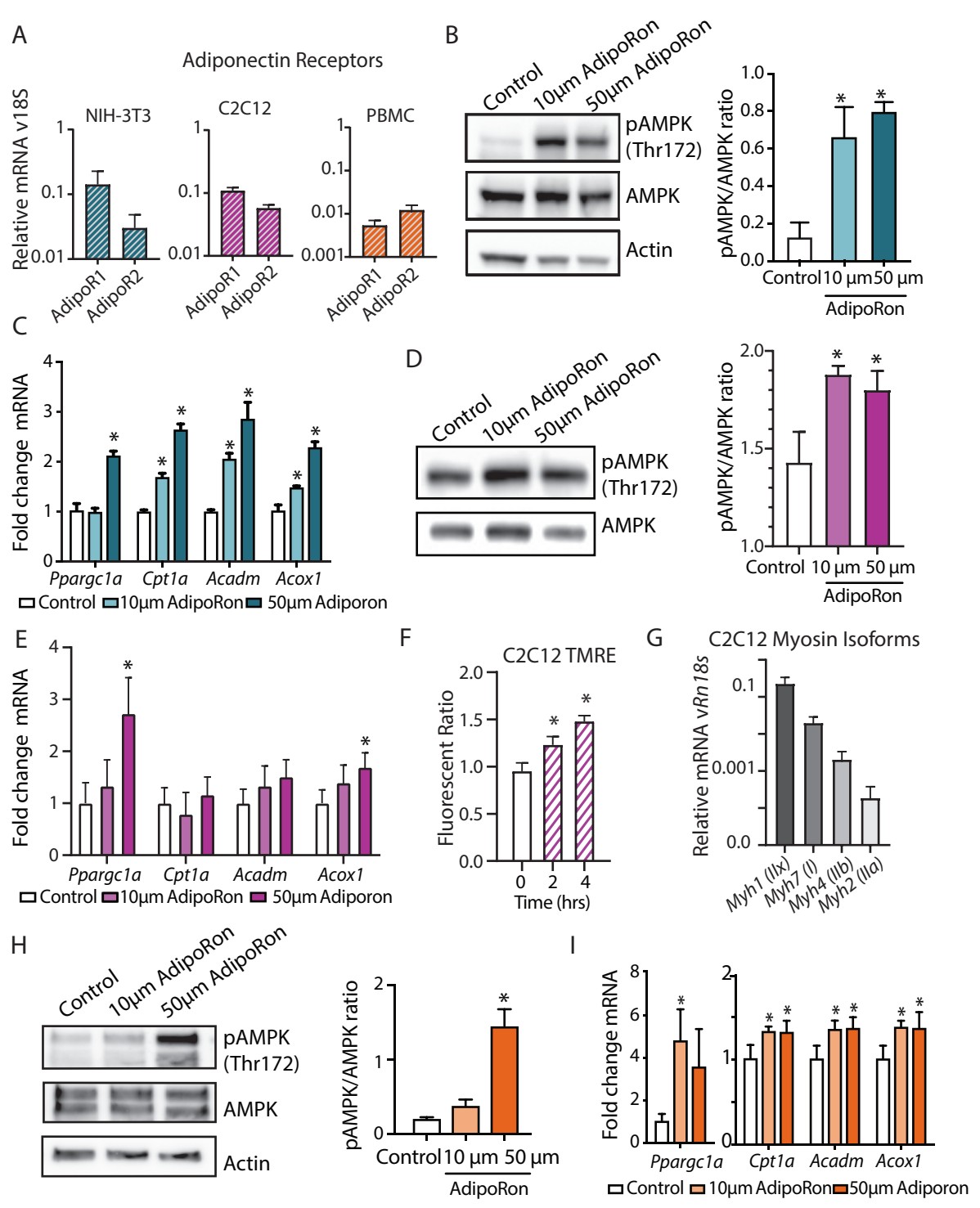

**Figure 6.** AdipoRon activates AMPK, increases the expression of *Ppargc1a*, and activates PGC-1a gene targets in diverse cultured cells. (**A**) RT-PCR detection of adiponectin receptors in NIH-3T3 fibroblast, day 7 differentiated C2C12 myotubes, and rhesus monkey peripheral blood mononuclear cells (PBMCs) (n=7, 7, and 4, respectively), (**B**) phosphorylation of AMPK at Thr172 after AdipoRon treatment (10 or 50 μm) for 10 min (n=3 per group) in NIH-3T3 fibroblasts, (**C**) transcript levels of *Ppargc1a* and its gene targets after 90 min of AdipoRon treatment (10 or 50 μm, n=4–5 per group, relative to *Rn18s* RNA), (**D**) phosphorylation of AMPK at Thr172 after AdipoRon treatment for 10 min in day 7 differentiated myotubes (n=3 per group), (**E**) transcript levels of *Ppargc1a* and its gene targets after 90 min of AdipoRon treatment (10 or 50 μm, n=7 per group, relative to *Rn18s* RNA), (**F**) mitochondrial membrane potential measured by TMRE assay at 2 and 4 hr of AdipoRon treatment (n=7 per group), (**G**) RT-PCR detection of myosin isoforms relative to *Rn18s* in day 7 differentiated myotubes, (**H**) phosphorylation of AMPK at Thr172 after AdipoRon treatment (10 or 50 μm) for 10 min in cultured rhesus

*Figure 6 continued on next page*

*Figure 6 continued*

monkey PBMCs (n=3–4 per group), and (**I**) transcript levels of *Ppargc1a* and PGC1a gene targets after 60 min of AdipoRon treatment (n=3–4 per group). Data shown as average ± SEM (*p<0.05).

distinct structural, biochemical, and metabolic properties. Studies in humans and nonhuman primates have previously demonstrated that there is a selective decrease in fiber size and mitochondrial activity of Type II fibers with aging (*Brunner et al., 2007*; *Murgia et al., 2017*; *Pugh et al., 2013*). AdipoRon treatment increased mitochondrial activity in a fiber-type-specific manner in the gastrocnemius muscle of aged mice and resulted in an increase in the number of Type IIa fibers. Given the narrow time frame of the intervention and prior links between PGC-1a expression and fiber type switching (*Lin et al., 2002*), it seems more likely that AdipoRon remodels existing fibers rather recruiting new fibers from the satellite pool. In situ histochemical staining for cytochrome c oxidase activity combined with fiber typing in muscle cryosections allowed us to investigate the structural and metabolic effects of AdipoRon based on fiber types. One of the striking findings of our study is that AdipoRon distinctively augmented mitochondrial metabolism in Type IIb fibers in aged mice. At the whole tissue level, this effect translated to increases in the force of contractility and endurance in the EDL, a glycolytic muscle that is rich in Type IIb fibers but not in the soleus muscle that is rich in Type I fibers.

Consistent with previously published studies (*Okada-Iwabu et al., 2013*), AdipoRon had a positive impact on glucoregulatory function in aged mice. In young, fit mice, the effects of AdipoRon at the cellular level were conserved but the systemic glucoregulatory and functional outcomes were not. It seems likely that AdipoRon may not be effective to stimulate muscle function outside of an existing age-related, genetic, or diet-induced decline. Experiments in isolated muscle preparations may shed light on the tissue autonomous versus systemic effects of AdipoRon on skeletal muscle metabolism, particularly in the context of aging. In our study, the beneficial impact of AdipoRon treatment in skeletal muscle was mechanistically linked with PGC1a activation. Previous studies on muscle-specific overexpression of PGC1a have reported similar findings; however, in some genetic models, whole-body insulin sensitivity was adversely affected (*Benton et al., 2008*; *Choi et al., 2008*; *Wong et al., 2015*). Our results show that AdipoRon treatment resulted in increase in the expression of endogenous *Ppargc1a* mRNA with concomitant modest but significant increases in PGC-1a protein levels. These data suggest that the activation of regulatory pathways upstream of PGC-1a may be key to exerting beneficial effects beyond skeletal muscle energetics, although we cannot rule out systemic effects of AdipoRon to alter metabolism in other tissues playing a role in glucoregulatory outcomes.

A decline in muscle energy metabolism precedes the onset of sarcopenia in nonhuman primates (*Pugh et al., 2013*), a finding that promoted this study of AdipoRon as a potential therapeutic agent. Our data show not only the activation of AMPK and of PGC-1a, but also reveal functional metabolic changes in response to AdipoRon treatment, in tissues as described above but also in cultured cells. In NIH-3T3 fibroblasts, mitochondrial membrane potential, cellular respiration, proliferation, and mitochondrial architecture were all impacted by AdipoRon. AMPK plays a central role in adaptive energetics and connects with other pathways regulating growth, including mTOR and insulin signaling (*Liu and Sabatini, 2020*). A broader investigation not limited to factors directly implicated in adiponectin receptor signaling could be fruitful and may reveal further connectivity among growth and energetic pathways.

Prior studies in renal endothelial cells report lipid remodeling subsequent to adiponectin receptor activation (*Choi et al., 2018*). It will be interesting to determine if the changes in mitochondrial function reported here extend to lipid metabolism and the composition of lipid structures including droplets, organelle membranes, and plasma membrane. Genetic studies of modest PGC-1a overexpression in preadipocytes identified similar changes in mitochondrial membrane potential and respiration that were coincident with greater capacity for lipid fuel utilization, changes in lipid handling, lipid storage, and membrane lipid composition (*Miller et al., 2019b*). Anti-inflammatory actions of AdipoRon could also exert beneficial muscle outcomes as recently reported in a genetic mice model of Duchenne muscular dystrophy and warrants investigation in our aging study (*Abou-Samra et al., 2020*). The impact of AdipoRon on differentiated C2C12 cells was similar but not equivalent to the response in NIH-3T3 fibroblasts. AMPK activating phosphorylation was increased, mitochondrial membrane potential was elevated, and *Ppargc1a* expression was increased; however, activation of gene targets of PGC-1a was not the same between the two cell types. Differences in relative levels of

AdipoR1 and AdipoR2 may account for cell type specificity in AdipoRon's actions. Furthermore, the expression of myosin isoforms is linked to metabolic status in muscles. It will be interesting to understand the expression of myosin isoforms among the individual myotubes, how that relates to cellular metabolism, and if that might account for differences in the ability of AdipoRon to influence gene targets of PGC-1a. In nonhuman primate PBMCs, AdipoRon is equally potent in activating AMPK and PGC1a indicating that AdipoRon's ability to stimulate the AMPK/PGC-1a axis may be translatable to nonhuman primates and perhaps humans. In addition, activation of adiponectin signaling has implications for aging and longevity as it shares signaling mechanisms involving AMPK and PGC1a with a known anti-aging intervention, calorie restriction (CR) (*Balasubramanian et al., 2017*). CR is associated with higher circulating adiponectin, especially the high molecular weight (HMW) isoform (*Miller et al., 2017*), which is the most potent in insulin sensitization.

Although these findings are promising, there are important caveats. The first is that the data shown here were generated in male mice only. Sex dimorphism is increasingly a focus in aging studies and across biomedical research (*Sampathkumar et al., 2020*). Our ongoing studies in mice and in nonhuman primates point to sex dimorphism in skeletal muscle mass and function, the impact of age on both parameters, and in the molecular and cellular effects of aging and CR. It will be critical to extend these studies of AdipoRon to female mice to determine its potential as an intervention to prevent and treat sarcopenia in both sexes. Additional studies translating the work to nonhuman primates, including both male and female monkeys, would set the stage for clinical investigations. The second limitation is that for logistical reasons, the genetic background of the young and old cohorts of mice was not the same in this study. It is possible that differences in underlying genetics contributed to differences in the outcome of AdipoRon treatment; however, a more likely explanation is that young mice are refractory to enhancement because they are already fit and healthy and the axis targeted by AdipoRon cannot be further boosted beyond this youthful setting. Finally, this study is cross-sectional, tested only one dose of AdipoRon, and investigated one time point only. A larger-scale study involving multiple time points with longitudinal measures would clarify when the optimum time for treatment occurs and would further delineate the age-related phenotypes that are responsive to stimulation of the adiponectin receptor signaling pathway. Taken together, our data suggest that AdipoRon, through its actions on mitochondrial metabolism, skeletal muscle composition, fiber fatigability and function, and physical capacity, is a promising therapeutic agent to preserve skeletal muscle function in aged populations.

## Materials and methods

### Mice and treatment

Two cohorts of mice were used. Six-week-old male B6C3F1 hybrid mice were obtained from Jackson Laboratories (Bar Harbor, ME) and 24-month-old male C57BL/6J mice from the NIA aged mouse colony. The animals were housed under controlled pathogen-free conditions. The animals were fed ad-libitum Purina LabChow 5001 diet with free access to water and placed on a 12-h light/dark cycle. AdipoRon (10 mg) (Adipogen, CA) was reconstituted with 500 μl of DMSO (stock) and stored at –20°C. For the acute experiment, the animals were given either a single intravenous dose of AdipoRon @1.2 mg/kg BW in PBS (n=5) or an equivalent volume of DMSO in PBS (n=3, Controls) in a total volume of 100 μl. EDL and soleus muscle groups were harvested after 90 min. For the chronic experiments in young B6C3F1 mice (n=12) and old C57BL/6J mice (n=20), the animals in each age group were divided into two groups and received either AdipoRon @1.2 mg/kg BW in PBS or an equivalent volume of DMSO in PBS via tail vein injections three times per week (Monday, Wednesday, and Friday) for 6 weeks. Body weight and food intake measurements were recorded once a week. Body composition was measured at the beginning and at the end of the treatment period using EchoMRI Body Composition Analyzer (Houston, TX). Rotarod balance measures were performed on a Columbus Instruments Rotamax system. During the test, the instrument speed increased from 5 to 60 rpm over a 3-min interval. Animals were trained on the instrument 1 day prior to the test and were allowed three attempts during test. Time on the rotarod and the speed when the mice falls are recorded. At the end of the treatment period, animals were sacrificed. EDL, soleus, and gastrocnemius muscle specimens were mounted in OCT oriented to present fiber cross-section for tissue sectioning or snap-frozen in

liquid nitrogen and stored at –80°C. All animal protocols were approved by the Institutional Animal Care and Use Committee at the William S. Middleton Memorial Veterans Hospital.

## Fasting glucose and insulin measurements

Overnight fasting blood samples were collected for measures of glucose (OneTouch Ultra Blue glucometer), and insulin levels (Ultra-Sensitive Mouse Insulin ELISA Kit (CystalChem, IL)). HOMA-IR (homeostasis model assessment of insulin resistance) index was calculated as [fasting serum glucose×fasting serum insulin/22.5].

## Metabolic chambers

Metabolic parameters and activity were measured (Columbus Instruments Oxymax/CLAMS metabolic chamber system (Columbus, OH)) in mice acclimated to housing in the chamber for approximately 24 hr, followed by 24 hr continuous data collection period.

## Ex vivo muscle force measurements

EDL and soleus hindlimb muscles were dissected and perfused with oxygenated (95% $O_2$, 5% $CO_2$) Tyrode's solution at room temperature (RT; NaCl 145 mM, KCl 5 mM, $CaCl_2$ 2 mM, $MgCl_2$ 0.5 mM, $NaH_2PO4$ 0.4 mM, $NaHCO_3$ 24 mM, EDTA 0.1 mM, and Glucose 10 mM). Isolated muscles were attached to a contractile apparatus capable of measuring force (Aurora Scientific) and were electrically stimulated using parallel platinum electrodes. Maximal twitch force and tetanic force were determined by adjusting the length of the muscle until maximal twitch force is reached, defining the optimal length. A 15-min period of rest allowed muscle to equilibrate to the new environment. Fatigability was defined as the decline in tetanic force following 10 min of continuous stimulation. Muscles were tetanically stimulated at 100 Hz for 500 ms every 5 se at a voltage that generates the maximal force. Following 10 min of fatiguing stimulation, a 20-min recovery period was allowed, where there was no stimulation in an oxygenated Tyrode's perfusion. After the recovery period, muscle was electrically stimulated again to quantify the amount of recovery force. A recovery force higher relative to fatiguing force was taken as an indication that the decline in force during fatiguing stimulation is reversible. The amount of fatigue and recovery force was represented as a percentage of the initial force (before fatiguing stimulation). Following these measurements, fatigued muscles were weighed, quick-frozen in liquid nitrogen, and stored at –80°C.

## Western blotting

Equal amounts of skeletal muscle protein extract (45 µg) were separated on Mini-Protean TGX precast protein gels (Bio-Rad, CA) and transferred to a PVDF membrane using a Trans-Blot semi-dry transfer system (Bio-Rad, CA). The membranes were blocked using 5% BSA in TBST for phospho-antibodies or 5% non-fat milk for other antibodies for 1 hr at RT. The following primary antibodies were used overnight: pAMPK Thr172 (Cell Signaling Technology #2535, Beverly, MA), AMPK (Cell Signaling Technology #2532), GAPDH (Cell Signaling Technology #2118), PGC1a (H300, Santa Cruz Biotechnology #sc-13067), and Beta-Actin (Sigma-Aldrich, #A1978), washed in TBST, and incubated with the respective HRP-conjugated secondary antibodies (Vector Labs, #PI:1000 and #PI:2000, Burlingame, CA) for 1 hr at RT. Proteins were detected (Supersignal West Pico or Femto Chemiluminescent substrate solutions [Thermo Fisher Scientific]) and digital images acquired (GE ImageQuant Gel Doc Imaging system). Densitometric analysis was carried out using Fiji software.

## Real-time PCR

RNA was extracted in TRIzol reagent and isolated (Direct-zol RNA Miniprep kit [ZymoResearch, Irvine, CA]). cDNA was synthesized from 1 µg of RNA (High-Capacity cDNA Reverse Transcription Kit [Applied Biosystems]). Real-time PCR reactions were carried out using iTaq Universal SYBR Green dye mix (Bio-Rad) with 10 ng cDNA per reaction. The treatment effect was analyzed by $2^{\Delta\Delta Ct}$ method (*Livak and Schmittgen, 2001*) and presented either as 'relative expression' compared to *Rn18s* mRNA, or as 'fold change' where relative expression of control samples was normalized to 1. Primer sequences for measured transcripts may be found in the supplementary info (*Supplementary file 1* Table 1).

## Histology

Serial OCT mounted cryostat sections (10 μm in thickness) were cut at –14°C (Leica Cryostat (Fisher Supply, Waltham, MA)). Freshly cut sections were stained for cytochrome c oxidase activity and digital images were captured the same day as described previously (*Pugh et al., 2013*). Briefly, sections were air-dried at RT for 10–15 min, incubated in a solution of 0.1 M phosphate buffer, pH 7.6, 0.5 mg/ml DAB (3,3′-diaminobenzidine), 1 mg/ml cytochrome c, and 2 μg/ml catalase at RT for ~30 min, washed with PBS, dehydrated, cleared and mounted under a glass coverslip (Permount; Thermo Fisher Scientific). Similarly for succinate dehydrogenase activity staining, the frozen muscle sections were air-dried for 10–15 min and incubated in SDH reaction mixture containing 1.5 mM nitroblue tetrazolium (NBT) and 50 mM di-sodium succinate in 0.2 M PBS (pH 7.6) at RT for ~20 minutes, washed with PBS, dehydrated, cleared, and mounted under a coverslip with Permount (*Punsoni et al., 2017*). Muscle fibrosis was determined by trichrome staining (Abcam, MA). For stain intensity and/or stain area analysis in digital images (n=5 per tissue), see below.

## Immunofluorescence

Fiber types (I, IIa, IIb, and IIx) were quantified in gastrocnemius muscle. Sections were air-dried for 30 min and then rehydrated with PBS for 5 min. Sections were then blocked with 0.5% BSA 0.5% Triton X-100 in PBS for 30 min at RT, then incubated with a cocktail of primary antibodies purchased from Developmental Studies Hybridoma Bank (DSHB, University of Iowa): BA-D5 (IgG2b, supernatant, 1:100 dilution) specific for MyHC-I, SC-71 (IgG1, supernatant, 1:100 dilution) specific for MyHC-IIa and BF-F3 (IgM, supernatant, 1:7.5 dilution) specific for MyHC-IIb for 1 hr at RT. After three washes with PBS (5 min each), the sections were incubated with secondary antibody cocktail (Invitrogen) to selectively bind to each primary antibody: goat anti-mouse IgG1 conjugated with Alexa Fluor488 (Invitrogen, #A21121); goat anti-mouse IgG2b conjugated with Alexa Fluor350 (Invitrogen, #A21140); goat anti-mouse IgM conjugated with Alexa Fluor594 (Invitrogen, #A211044) for 1 hr in the dark at RT. After three washes with PBS (5 min each) and a brief rinse in water, the sections were mounted in 85% glycerol in PBS for imaging. Type IIa fibers will appear green, IIb as red, I as blue, and the fibers that are not stained by these antibodies will appear black and are classified as Type IIx.

## Digital image capture and analysis

Digital images were captured using a 20× objective in a Leica DM4000B microscope equipped with Retiga 4000R digital camera (QImaging Systems, Surrey, BC). Background correction was conducted for all images using an unstained adjacent area of the slide. Images were converted to 8-bit format and inverted. Fiber type was classified based on immunostaining for myosin isoform and individual fibers were outlined using a freehand tool and the cross-sectional area. Stain intensity of the cytochrome c oxidase and succinate dehydrogenase was quantified using ImageJ/Fiji software. Using a custom-generated algorithm, the blue stained areas for collagen were separated and highlighted from the rest of the image and % stained area was quantified (MIPAR software). Mitochondrial analysis was performed on cells in ImageJ by applying image deconvolution, background subtraction, adaptive binarization, and segmentation algorithms, followed by particle analysis and morphology analysis with the ImageJ plugin MiNA (*Valente et al., 2017*) to quantify intensity and mitochondrial branching.

## Cell culture and reagents

NIH-3T3 fibroblasts were purchased (ATCC; CRL-1658) and cultured in DMEM supplemented with 10% bovine serum and 1% penicillin/streptomycin. The cells were treated with vehicle (DMSO in media) or AdipoRon (10 or 50 μM in media) for 10 (AMPK phosphorylation) or 60 min (gene expression analysis). After treatment, the cells were washed once with PBS and then lysed in TRIzol or RIPA buffer depending on the experiment. C2C12 cells were purchased (ATCC; CRL-1772) and grown in 10% bovine serum and 1% penicillin/streptomycin. Differentiation was initiated after 2–3 days growth by exposure to differentiation media: Serum-free DMEM supplemented with 2% equine serum, 1% penicillin/streptomycin, and 172 nM insulin. Myoblasts were fed differentiation media every 24 hr for 5–7 days until myotube morphology was reached. All cell lines were authenticated by STR profiling and cleared of mycoplasma contamination.

## TMRE and oxygen consumption assays

Cells were treated with vehicle or AdipoRon (50 µM) for 2 or 4 hr. For TMRE fluorescence, cells were seeded at $5×10^4$ overnight and treated with AdipoRon (50 µm) for 2 or 4 hr. Cells were then incubated for 30 min in 100 nM TMRE dye in media and, after equilibrating for 10 min, plates were read at 530 nm excitation/580 nm emission, parallel treatment with 10 µM uncoupler FCCP (Cayman Chemical) accounted for mitochondrial oxidative phosphorylation. Data shown as fluorescent ratio of given treatment over uncoupled control. For oxygen consumption assays, cells were trypsinized and equal numbers ($4×10^5$) were resuspended in respiration buffer and loaded on to the PreSens Oxoplates (Regensburg, Germany). Appropriate ambient and anoxic controls were included in the same plate.

## ATP luminescence assay

Relative ATP levels were quantified ATPLite assay (Perkin-Elmer, Waltham, MA; INFINITE M1000 PRO microplate reader (TECAN, Grodig, Austria)) in cells cultured as above and treated with vehicle or AdipoRon (50 µM) for the prescribed times.

## Cell proliferation assay

NIH-3T3 cell proliferation was quantified according to the CyQUANT Cell Proliferation Assay Kit (Molecular Probes, Inc, Eugene, OR) at four cell densities 1600–6700 cells/well. In brief, cells were seeded in microplate wells in growth medium at desired densities along with serial dilutions of cells for determination of a cell number standard curve. Cells were incubated for 24 hr after which culture medium was removed, and the number of cells quantified according to the manufacturer's instructions.

## Nonhuman primate PBMC isolation and culture

Blood was collected from primates at the Wisconsin National Primate Research Center (WNPRC) with the approval of the Institutional Animal Care and Use Committee of the Office of the Vice Chancellor for Research and Graduate Education of the University of Wisconsin, Madison. PBMCs were isolated from 6 ml of blood using SepMate-50 tubes (Stemcell Technologies, Cambridge, MA). Isolated primate PBMC's ($1×10^6$) were cultured in 10 cm plates in DMEM supplemented with 10% fetal bovine serum and 1% penicillin/streptomycin. Cells were treated with vehicle (DMSO in media) or AdipoRon (10 or 50 µM in media) for 5 (AMPK phosphorylation) or 60 min (gene expression analysis). PBMCs were collected by centrifugation and the cell pellet was washed and lysed in TRIzol or RIPA buffer.

## Statistical analysis

Results are expressed as mean ±pooled SEM. For cell culture and animal experiments, data sets involving two groups were analyzed by unpaired two-tailed t-tests and for data sets with three groups one-way ANOVA followed by Bonferroni's multiple comparison test with a cutoff of $p < 0.05$.

# Acknowledgements

The authors would like to acknowledge support from the Department for Veterans Affairs VA Merit BX003846, BX004031, NIH/NIA AG040178, AG056771, NIH training fellowships AG000213 (AES) and GM083252 (PRH), and the Glenn Foundation for Medical Research. This publication was made possible in part by NIH/ORIP grant P51OD011106 to the Wisconsin National Primate Research Center, University of Wisconsin-Madison. This work was supported by the use of facilities and resources at the William S Middleton Memorial Veterans Hospital, Madison, WI.

# Additional information

### Funding

| Funder | Grant reference number | Author |
| --- | --- | --- |
| U.S. Department of Veterans Affairs | BX003846 | Priya Balasubramanian Anne E Schaar Rozalyn M Anderson |

| Funder | Grant reference number | Author |
|---|---|---|
| U.S. Department of Veterans Affairs | BX004031 | Dudley W Lamming |
| National Institute on Aging | AG040178 | Alex B Smith<br>Scott Baum<br>Ricki J Colman<br>Rozalyn M Anderson |
| National Institute on Aging | AG056771 | Dudley W Lamming |
| National Institutes of Health | AG000213 | Anne E Schaar |
| National Institutes of Health | GM083252 | Porsha R Howell |
| National Institutes of Health | P51OD011106 | Scott Baum<br>Ricki J Colman |
| Glenn Foundation for Medical Research | | Rozalyn M Anderson |

The funders had no role in study design, data collection and interpretation, or the decision to submit the work for publication.

### Author contributions

Priya Balasubramanian, Anne E Schaar, Conceptualization, Investigation, Writing – review and editing; Grace E Gustafson, Alex B Smith, Investigation, Visualization; Porsha R Howell, Data curation; Angela Greenman, Funding acquisition, Investigation; Scott Baum, Ricki J Colman, Gary M Diffee, Investigation, Resources; Dudley W Lamming, Resources; Rozalyn M Anderson, Conceptualization, Funding acquisition, Writing - original draft

### Author ORCIDs

Priya Balasubramanian http://orcid.org/0000-0003-0912-5363
Grace E Gustafson http://orcid.org/0000-0001-8343-4493
Alex B Smith http://orcid.org/0000-0002-2615-8899
Dudley W Lamming http://orcid.org/0000-0002-0079-4467
Rozalyn M Anderson http://orcid.org/0000-0002-0864-7998

### Ethics

This study was performed in strict accordance with the recommendations in the Guide for the Care and Use of Laboratory Animals of the National Institutes of Health to minimize animal harm and suffering. All animal protocols were approved by the Institutional Animal Care and Use Committee at the University of Wisconsin, Madison on animal protocols #RA-0005-1 and #RA-0007-1.

### Decision letter and Author response

Decision letter https://doi.org/10.7554/eLife.71282.sa1
Author response https://doi.org/10.7554/eLife.71282.sa2

---

## Additional files

### Supplementary files

• Supplementary file 1. RT-PCR forward and reverse primer sequences and corresponding genes used for identifying transcripts in mouse tissue and diverse cell types.

• Transparent reporting form

### Data availability

All data generated or analyzed during this experimental study are included in the manuscript and supporting files used to generate figures 1-6 have been deposited to Dryad at URL (https://doi.org/10.5061/dryad.x95x69pkt).

The following dataset was generated:

| Author(s) | Year | Dataset title | Dataset URL | Database and Identifier |
|---|---|---|---|---|
| Anderson RM | 2022 | Data from: Adiponectin receptor agonist AdipoRon improves skeletal muscle function in aged mice | https://dx.doi.org/10.5061/dryad.x95x69pkt | Dryad Digital Repository, 10.5061/dryad.x95x69pkt |

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
