## [Editor Report]

In this manuscript, the authors provide promising results for the treatment of age-related sarcopenia with AdipoRon, a drug that targets the receptors for adiponectin. This is a well done study using an agonist (AdipoRon) involved in lipid and mitochondrial metabolism regulation to mitigate age-related muscle loss in mice.

---

## [Decision Letter]

**Decision letter after peer review:**

Thank you for submitting your article "Adiponectin receptor agonist AdipoRon improves skeletal muscle function in aged mice" for consideration by *eLife*. Your article has been reviewed by 3 peer reviewers, and the evaluation has been overseen by a Reviewing Editor and Carlos Isales as the Senior Editor. The following individual involved in review of your submission has agreed to reveal their identity: John C Newman (Reviewer #1).

Essential revisions:

1) Use an additional, more reliable potentiometric dye (e.g. TMRE) instead to validate the JC-1 findings

2) Justify the different time points or present time-course experiments for the cell culture results, possibly in a more relevant cell model. The use of 3T3 and PBMC instead of more relevant cell models

3) Better description of the sample size, especially for endpoint measurements.

Describe in more details the sample sizes for each cohort

4) Re-evaluate the claims of improved muscle function and increased lipid metabolism.

Optional – do the results hold true in the opposite sex, though I can understand if budgetary constraints prevent the authors for repeating the assays in female mice.

*Reviewer #1 (Recommendations for the authors):*

1) Overall this work provides a strong justification for continuing studies of AdipoRon in larger rodent experiments and in non-human primates, and provides excellent biomarker candidates for those. The underpowered mouse cohorts, though, leads to a lack of whole-animal muscle functional data. "Improved skeletal muscle function" and the like, used throughout, could be read to imply an improvement in relevant behavioral outcomes (e.g. rotatod, grip strength, frailty index, running speed, spontaneous activity, etc.) which are not present here. Muscle mass does not increase in the aged mice either. This is the key issue to be careful with in the presentation of the findings.

2) What were the final numbers in the aged cohorts? Methods states starting with 20 mice total, and Figure 1 legend says ending with 7-10 control and 6-8 AdipoRon. Did some mice die or were otherwise excluded? Were more of these in the AdipoRon group? One way or another, it is important to account for missing data and any signals of safety issues.

3) I appreciate the prominent mentions of the limitations of a male-only study, although an even better approach would be to validate some of the findings in a female cohort! Given the pilot-sized experiments this seems not entirely unreasonable.

4) As in the public review, why use different strains for young and old mice? I assume a practical reason (young hybrids readily available on site, old B6 available from NIA) which is understandable enough but the limitations of cross-comparing different strains at different ages should be explicitly stated.

5) As in the public review, why use NIH3T3 for in vitro pathway studies instead of a more muscle-like cell line or primary cell type? Similarly for using primate PBMCs. I didn't see confirmation that these cell types express adiponectin receptors, or other downstream components, and how similar the molecular responses are expected to be to muscle. Is there literature about PMBCs responding to adiponectin physiologically in vivo?

6) The densitometry for Pgc1a in Figure 1I does not obviously correspond to the Pgc1a band intensity, and if normalized to Ponceau (which is not stated) the Ponceau difference between groups doesn't obviously account for a 2x difference in Pgc1a/Ponceau. This is worth a double-check.

7) It is not always obvious what the reference condition is for presentation of "relative mRNA" and "mRNA Fold change" panels – one expects normalization to a comparison group mean set to 1.0 for each panel, but for most that comparison group is not clear.

8) The in vitro acute treatments included varying treatment times, which were not always specifically described for each experiment in the legend or methods.

9) The text states that food consumption was the same between groups in the aged mouse cohort, but these data do not appear to be shown. If food consumption is the same, activity is the same, and energy expenditure is the same, what explains the fat mass weight loss in the AdipoRon-treated group? Given the changes observed in fat metabolism and the loss of fat mass in old mice, why was no difference observed in RER?

10) Why are the phenotype data that were collected different between the young and aged chronic treatment cohorts (Figure 1 and 3)? E.g. no body composition measurements, insulin, HOMA-IR, metabolic cages in young mice.

11) A brief explanation of mTOR inhibition by AMPK activation (cell proliferation) could be included in the discussion of this study to further explain sarcopenia and other age-related muscle loss.

12) The explanation of the glucoregulatory function in aged mice by AdipoRon could be accompanied by the catabolic pathways triggered by AMPK activation.

13) The first two-thirds of the section titled "nonhuman primate cultured cells" continues to describe experiments in NIH3T3, which is confusing. The first sentence of the Figure 6 legend causes the same confusion, although the panel descriptions are clear which is which.

14) In general, I appreciate the careful language used to describe suggestive but not statistically significant results.

15) How was AdipoRon obtained? Was it donated by the company? Were there any considerations provided by the company for the study?

*Reviewer #2 (Recommendations for the authors):*

With the exception of the JC-1 assays, the study appears well done, with sufficient rigor in terms of numbers of animals used per group etc. However, in addition to the well known problems with the JC-1 dye being used to measure mitochondrial membrane potential, there a couple of minor quirks, which can easily be corrected.

1) On page 4, references are made to Benton and Ruas as showing PGC-1a over expression protecting against muscle aging in mice. Neither of these papers show this. Benton et al. has no aging data. Ruas also has no aging data, and is in rats, not mice. Cite the correct references, and remove these.

2) Rename the EchoMRI to "body composition measures". This is super confusing to those who don't know this instrument (it is neither "echo" nor "MRI").

3) On page 5 there is reference to EDL and soleus being glycolytic and oxidative. As the authors point out repeatedly later on, these muscle are mixed fibre types, albeit heavily glycolytic or oxidative. I think there should be consistency on this point.

Regarding JC-1, this results should be removed altogether for measuring mitochondrial membrane potential. Its problems have been documented extensively, and I'm not sure why investigators still use it. Its well known to be non specific to plasma membrane potential in addition to mitochondria, and has multiple problems with regards to specificity. The authors should revise their manuscript substantially on this point. Use of this assay spoils an otherwise good paper.

*Reviewer #3 (Recommendations for the authors):*

Page 6: "Expression levels of both receptors were numerically lower in AdipoRon treated mice. In soleus muscle, AdipoR1 expression was not altered with age (Figure S2) and AdipoRon increased expression, although neither change reached significance. AdipoR2 levels declined with age in soleus of control mice, but not in mice treated with AdipoRon."

The two sentences seem to be at odds with one another. The first sentence is also not consistent with the data in figure S2.

Page 9: JC-1 measurement of mitochondrial membrane potential can give artifacts, especially when fluorescence is measured via a plate reader and not visualized via microscopy or flow cytometry. I would encourage the authors to validate their results with TMRE or TMRM staining.

Page 9: Why the different time points for measurements of cell proliferation, markers of mitochondrial function, and ATP levels (24 vs 48h vs 12h) ?

Page 11: :The inference is a shift toward fatty acid fuel utilization, aligning with prior reports of lipid remodeling outcomes reported in renal endothelial cells"

I don't know whether this inference is warranted without RER measurements or at least evidence of activation of fitty acid oxidation. Indeed, the in vivo indirect calorimetry failed to show a shift towards lipid catabolism, which is at odds with this statement.

Page 12: Could you provide a justification for using only male animals and using two different strains between old and young animals?

Page 13: Please include clone or catalog number for all antibodies used.

---

## [Author Response]

Essential revisions:1) Use an additional, more reliable potentiometric dye (e.g. TMRE) instead to validate the JC-1 findings

We have repeated the experiments on mitochondrial membrane potential using the TMRE dye as suggested. The outcome of increased membrane potential in response to AdipoRon treatment first detected using JC-1 is now validated with the new approach using TMRE and the data are shown in the revised Figure 5. We have also validated this outcome in an additional cell type (differentiated C2C12) now shown in the revised Figure 6.

2) Justify the different time points or present time-course experiments for the cell culture results, possibly in a more relevant cell model. The use of 3T3 and PBMC instead of more relevant cell models

We have conducted parallel experiments in differentiated C2C12 myotubes and confirm that AdipoRon activates expression of PGC-1a and its targets in addition to increasing membrane potential as measured by TMRE. We have revised text to justify the selection of time points for the various measured parameters.

3) Better description of the sample size, especially for endpoint measurements.Describe in more details the sample sizes for each cohort

We apologize that this was not clearer before, we have revised the manuscript accordingly.

4) Re-evaluate the claims of improved muscle function and increased lipid metabolism.

We have toned down the language here, and throughout make a clear distinction between statements referring to the data shown and those that are inference or speculation on our part.

Optional – do the results hold true in the opposite sex, though I can understand if budgetary constraints prevent the authors for repeating the assays in female mice.

We have just completed a follow up study using oral delivery of AdipoRon for a longer duration (4 months) in males and females and can confirm that AdipoRon treatment is associated with improvements in indices of systemic metabolism in males and females. In this study AdipoRon is associated with significant protection against age-related functional loss in males, although the study is not yet complete and we expect longitudinal analysis will yield greater insight than a cross-sectional approach. We are currently assessing the tissue, cellular, and molecular effects of aging and of AdipoRon in both males and females.

Reviewer #1 (Recommendations for the authors):1) Overall this work provides a strong justification for continuing studies of AdipoRon in larger rodent experiments and in non-human primates, and provides excellent biomarker candidates for those. The underpowered mouse cohorts, though, leads to a lack of whole-animal muscle functional data. "Improved skeletal muscle function" and the like, used throughout, could be read to imply an improvement in relevant behavioral outcomes (e.g. rotatod, grip strength, frailty index, running speed, spontaneous activity, etc.) which are not present here. Muscle mass does not increase in the aged mice either. This is the key issue to be careful with in the presentation of the findings.

We agree with the reviewer on this point. In revising the manuscript, we have been careful to distinguish between statements of fact that those that are more interpretive or speculative. We recognize the limited set of parameters measured here, and as this reviewer will appreciate it is tricky with mice to get estimates of muscle mass in vivo – most investigations just report “lean mass” which is not muscle of course but everything that is not fat. As mentioned in our response above we have been conducting a follow-up study in mice that will allow for a more detailed investigation of the tissue, cellular, and molecular signatures of aging and of the response to AdipoRon.

2) What were the final numbers in the aged cohorts? Methods states starting with 20 mice total, and Figure 1 legend says ending with 7-10 control and 6-8 AdipoRon. Did some mice die or were otherwise excluded? Were more of these in the AdipoRon group? One way or another, it is important to account for missing data and any signals of safety issues.

We agree. In the revised figures we now specify the numbers of animals/observations contributing to the data reported. We lost 2 of the AdipoRon mice but given the advanced age of the cohort this was not unexpected. We were limited to the 20 starting mice and a loss of 2 (10%) in the period from 25 to 27 months of age (the study period) is aligned with the mortality expectation for this line for which 50% mortality is around 30 months of age.

3) I appreciate the prominent mentions of the limitations of a male-only study, although an even better approach would be to validate some of the findings in a female cohort! Given the pilot-sized experiments this seems not entirely unreasonable.

As mentioned above we have undertaken a follow up study and are analyzing the tissues now. The study described in this manuscript was exploratory and at the time we could only work with what was to hand. The follow up study is more comprehensive but we are eager to share our results now as we see terrific promise in the use of adiponectin receptor stimulation as a means to counter skeletal muscle aging. Our hope is that this paper will draw others to investigate metabolism as a target for delaying aging in skeletal muscle and hopefully in a shared effort we can move the field forward.

4) As in the public review, why use different strains for young and old mice? I assume a practical reason (young hybrids readily available on site, old B6 available from NIA) which is understandable enough but the limitations of cross-comparing different strains at different ages should be explicitly stated.

We realize the importance of genetics and highlight this limitation in the revised manuscript. We would add that our follow up study in C3B6F1 hybrid mice reproduces the outcomes described here. In those mice there was very little impact of AdipoRon in young mice that were fit and healthy (6 months of age), modest differences in metabolic and functional outcomes late middle age (22 months) and significant differences at advanced age (28 months).

5) As in the public review, why use NIH3T3 for in vitro pathway studies instead of a more muscle-like cell line or primary cell type? Similarly for using primate PBMCs. I didn't see confirmation that these cell types express adiponectin receptors, or other downstream components, and how similar the molecular responses are expected to be to muscle. Is there literature about PMBCs responding to adiponectin physiologically in vivo?

We have conducted analysis of AdipoR1 and AdipoR2 expression in each of the cultured cell models (NIH-3T3; C2C12; and PBMC), and these data are now shown in Figure 6. We do expect there to be cell type specificity in the outcome of AMPK/PGC-1a axis activation as this is a primary mode of action for adiponectin receptor stimulation; however, we suspect that there will be differences by cell type as far as the cellular consequence. Of relevance here, we expect differences by muscle group in the molecular consequence of AMPK/PGC-1a activation depending on fiber type distribution. Other skeletal muscle tissue resident cell types may also contribute. We know that age can impact tissue composition (fibrosis, adiposity, vascularization, infiltration of immune cells), and recognize the potential for differences in the responsiveness of various effectors both upstream and downstream in the signaling pathway. We should be able to make some headway in defining these differences in our ongoing follow up study.

6) The densitometry for Pgc1a in Figure 1I does not obviously correspond to the Pgc1a band intensity, and if normalized to Ponceau (which is not stated) the Ponceau difference between groups doesn't obviously account for a 2x difference in Pgc1a/Ponceau. This is worth a double-check.

We apologize for this error, there was a mistake in the quantification that I didn’t catch and I am relieved that you spotted this so we could fix it. We now show that the enrichment of PGC-1a protein is ~1.5 fold and state that the relative density based on normalization to ponceau, now shown in revised (Figure 1). As might be expected we do have some variability among individuals but this is a common feature in aging studies, in fact we believe that increased variance is a trait of aging and are pursuing this idea in our primate studies.

7) It is not always obvious what the reference condition is for presentation of "relative mRNA" and "mRNA Fold change" panels – one expects normalization to a comparison group mean set to 1.0 for each panel, but for most that comparison group is not clear.

We apologize for not being clear in our description. Briefly, there were two main ways in which the data were presented. The first, labeled “relative expression” shows the data as δ-CT against 18S, for this allows for comparison of expression among transcripts detected in the same specimen and is applied only for primer sets of equivalent efficiency. The second, labeled “fold change” normalizes the controls to 1 to emphasize the impact of AdipoRon. We now include this description in the methods and clarify in the text which comparisons are being made.

8) The in vitro acute treatments included varying treatment times, which were not always specifically described for each experiment in the legend or methods.

We apologize for the oversight, these details are now provided.

9) The text states that food consumption was the same between groups in the aged mouse cohort, but these data do not appear to be shown. If food consumption is the same, activity is the same, and energy expenditure is the same, what explains the fat mass weight loss in the AdipoRon-treated group? Given the changes observed in fat metabolism and the loss of fat mass in old mice, why was no difference observed in RER?

The reviewer raises important issues, a few explanations come to mind. Although we did not see differences in 12hour averages for EE or RER or activity, on closer inspection looking at 15-minute increments we do see changes indicative of differences between control and AdipoRon treated animals, but at specific time intervals only. These data are now shown in revised Figure 1. The magnitude of difference is modest; however, having the animals in the cages to acclimatize 24 hours before the start of measures means that they are at least a day out from the last injection with AdipoRon. In addition, the mice had continuous access to food such that circadian rhythm is quite blunted. In our experience, meal fed animals show much clearer profiles of fed and fasted but that was not the case with ad libitum feeding regimen used here. In hindsight it might have been worth conducting additional measures directly following injection and that is something we can consider doing in a separate group of mice. We would add confidentially that in our follow up study the trends reported here are reproduced, both males and females show changes in their RER and EE patterns, being quicker to adapt to feeding and to resolve post feeding.

10) Why are the phenotype data that were collected different between the young and aged chronic treatment cohorts (Figure 1 and 3)? E.g. no body composition measurements, insulin, HOMA-IR, metabolic cages in young mice.

Substantive changes in young animals were not expected, mice of 6 months of age were lean and fit. In our follow up study, we confirm body and HOMA-IR effects in old C3B6F1 hybrid mice. We now provide insulin data from the young mice on AdipoRon showing that insulin levels are not altered, based on the fact that the body weight did not differ we would argue a change in body fat is unlikely.

11) A brief explanation of mTOR inhibition by AMPK activation (cell proliferation) could be included in the discussion of this study to further explain sarcopenia and other age-related muscle loss.

Thank you for this great suggestion. We have added this to the discussion.

12) The explanation of the glucoregulatory function in aged mice by AdipoRon could be accompanied by the catabolic pathways triggered by AMPK activation.

This is also a great suggestion, we have added it to the discussion.

13) The first two-thirds of the section titled "nonhuman primate cultured cells" continues to describe experiments in NIH3T3, which is confusing. The first sentence of the Figure 6 legend causes the same confusion, although the panel descriptions are clear which is which.

Good point. We have rewritten this section addressing this concern and adding the new data on the differentiated C2C12 as shown in revised Figure 6.

14) In general, I appreciate the careful language used to describe suggestive but not statistically significant results.

Thanks, as mentioned above aging is accompanied by increase in variance for a host of measured parameters. We favor discussing all outcomes that we believe are biologically meaningful while being careful to note whether or not there was statistical significance in differences reported.

15) How was AdipoRon obtained? Was it donated by the company? Were there any considerations provided by the company for the study?

AdipoRon was obtained at considerable cost. As indicated in our declaration of no conflict, we did not receive favor or inducements of any kind from the manufacturer or any other party.

Reviewer #2 (Recommendations for the authors):With the exception of the JC-1 assays, the study appears well done, with sufficient rigor in terms of numbers of animals used per group etc. However, in addition to the well known problems with the JC-1 dye being used to measure mitochondrial membrane potential, there a couple of minor quirks, which can easily be corrected.

We have replaced the JC-1 assay with the TMRE as recommended. Although the outcome is similar, TMRE confirms the ability of AdipoRon to impact mitochondrial membrane potential, we are eager to make sure we are using the right tool for the job by eliminating the possibility that our signature is compromised by changes in extra-mitochondrial membranes. Thank you for bringing this to our attention.

1) On page 4, references are made to Benton and Ruas as showing PGC-1a over expression protecting against muscle aging in mice. Neither of these papers show this. Benton et al. has no aging data. Ruas also has no aging data, and is in rats, not mice. Cite the correct references, and remove these.

We have removed the incorrect references and replaced them with the correct ones.

2) Rename the EchoMRI to "body composition measures". This is super confusing to those who don't know this instrument (it is neither "echo" nor "MRI").

Thanks for pointing this out, we have made the suggested correction to our manuscript.

3) On page 5 there is reference to EDL and soleus being glycolytic and oxidative. As the authors point out repeatedly later on, these muscle are mixed fibre types, albeit heavily glycolytic or oxidative. I think there should be consistency on this point.

The reviewer is spot on here and we apologize for not being clearer about this in our initial submission. We have revised the manuscript to make sure we are consistent in our description of the fiber type and metabolic setting of the different muscle groups.

Reviewer #3 (Recommendations for the authors):Page 6: "Expression levels of both receptors were numerically lower in AdipoRon treated mice. In soleus muscle, AdipoR1 expression was not altered with age (Figure S2) and AdipoRon increased expression, although neither change reached significance. AdipoR2 levels declined with age in soleus of control mice, but not in mice treated with AdipoRon."The two sentences seem to be at odds with one another. The first sentence is also not consistent with the data in figure S2.

We are grateful to the reviewer for picking up on this clumsy wording. The first part is about gastrocnemius and the second refers to soleus. We have revised the text accordingly.

Page 9: JC-1 measurement of mitochondrial membrane potential can give artifacts, especially when fluorescence is measured via a plate reader and not visualized via microscopy or flow cytometry. I would encourage the authors to validate their results with TMRE or TMRM staining.

Thanks, we have replaced the JC-1 with TMRE as requested, please see our responses to reviewers 1 and 2 above.

Page 9: Why the different time points for measurements of cell proliferation, markers of mitochondrial function, and ATP levels (24 vs 48h vs 12h) ?

The timing of the experiments was matched to the outcomes being measured. For the immediate responses, cell signaling through AMPK and PGC-1a directed gene expression, we measured within 90 mintues based on time course experiments. The activation of AMPK is transient, as is the activation of gene expression. For physiological responses, membrane potential and respiration, we measured hours after exposure and show data from 2 and 4 hours. For ATP production we include data that we had left out initially. At 2 and 4 hours ATP production is lower but then corrects and is elevated relative to untreated cells at 12 hours. We have no explanation for the biphasic response which is why we had left it out of the initial submission. For proliferation we were guided by the doubling time of the cells, and for the and mitochondrial remodeling we were guided by a time course showing gradual changes that become significant at 24 hours and persist to 48 hours. We have included some of this detail in the revised manuscript and hope we have struck the correct balance with just enough information without getting lost in the detail.

Page 11: :The inference is a shift toward fatty acid fuel utilization, aligning with prior reports of lipid remodeling outcomes reported in renal endothelial cells" I don't know whether this inference is warranted without RER measurements or at least evidence of activation of fitty acid oxidation. Indeed, the in vivo indirect calorimetry failed to show a shift towards lipid catabolism, which is at odds with this statement.

We have rephrased this sentence to separate out what is known and reported previously in the literature from speculation on our part. The reviewer is correct that the definitive test would be done using ex vivo tissues to demonstrate preference for lipid fuel. We are pursuing this avenue now but in the meantime have made changes in the revised manuscript.

Page 12: Could you provide a justification for using only male animals and using two different strains between old and young animals?

At the time these experiments were conducted we were limited by what was available. We expanded our thoughts on these limitations in the discussion. Please also see the responses to reviewers 1 and 2 above.

Page 13: Please include clone or catalog number for all antibodies used.

Thank you for pointing out this oversight, this information is now provided.